# Total and endothelial cell-derived cell-free DNA in blood plasma does not change during menstruation

**Nicole Laurencia Yuwono**[ID]**, Claire Elizabeth Henry, Caroline Elizabeth Ford, Kristina Warton**[ID]*****

Gynaecological Cancer Research Group, Adult Cancer Program, Lowy Cancer Research Centre, Department of Obstetrics & Gynaecology, School of Women's and Children's Health, Faculty of Medicine, University of New South Wales, Sydney, NSW, Australia

* k.warton@unsw.edu.au

## Abstract

Assays measuring cell-free DNA (cfDNA) in blood have widespread potential in modern medicine. However, a comprehensive understanding of cfDNA dynamics in healthy individuals is required to assist in the design of assays that maximise the signal driven by pathological changes, while excluding fluctuations that are part of healthy physiological processes. The menstrual cycle involves major remodelling of endometrial tissue and associated apoptosis, yet there has been little investigation of the impact of the menstrual cycle on cfDNA levels. Paired plasma samples were collected from 40 healthy women on menstruating (M) and non-menstruating (NM) days of their cycle. We measured total cfDNA by targeting ALU repetitive sequences and measured endothelial-derived cfDNA by methylation-specific qPCR targeting an endothelium-unique unmethylated CDH5 DNA region. CfDNA integrity and endothelial cfDNA concentration, but not total cfDNA, are consistent across time between NM and M. No significant changes in total (ALU-115 p = 0.273; ALU-247 p = 0.385) or endothelial cell specific (p = 0.301) cfDNA were observed, leading to the conclusion that menstrual status at the time of diagnostic blood collection should not have a significant impact on the quantitation of total cfDNA and methylation-based cancer assays.

## Introduction

Cell-free DNA (cfDNA) in blood is a promising source of biomarkers for a range of conditions such as cancer detection and transplantation [1]. It is generally accepted that cfDNA is released into the blood due to cells dying via apoptosis and is also linked to inflammation [2]. cfDNA can be released into the blood in healthy individuals, patients with benign diseases and patients with cancer, and therefore it is important to distinguish between amounts of cfDNA from different cells of origin when developing cfDNA based biomarkers. For example there is a rise in pancreatic acinar and ductal cells-derived cfDNA in pancreatic cancer and pancreatic β-cell cfDNA in diabetes [3] as well as erythroid cfDNA in anaemia [4]. In healthy individuals, cfDNA has been shown to be predominantly of leukocyte origin however, other tissues such as the liver can also contribute to

**Data Availability Statement:** All relevant data are within the manuscript and its Supporting Information files.

**Funding:** Funding have been received from The Department of Education, Skills and Employment

(NLY) (https://www.education.gov.au/research-training-program), Beth Yarrow Memorial Award in Medical Science (NLY) (https://wch.med.unsw.edu.au/sites/default/files/UNSW%20Beth%20Yarrow%20Memorial%20Award%20in%20Medical%20Science%20-%20Guidelines_v1%2028FEB2019.pdf), Translational Cancer Research Network supported by the Cancer Institute NSW (NLY) (http://www.tcrn.unsw.edu.au/tcrn-grants), GO Research Fund (CEH), Ovarian Cancer Research Foundation (KW) (GA-2018-14) (https://www.ocrf.com.au/). The funders had no role in study design, data collection and analysis, decision to publish, or preparation of the manuscript.

**Competing interests:** I have read the journal's policy and the authors of this manuscript have the following competing interests: KW holds stock in Guardant Health, Exact Sciences and Epigenomics AG. No other authors have competing interests. This does not alter our adherence to PLOS ONE policies on sharing data and materials. There are no restrictions on sharing of data and/or materials from this publication.

the cfDNA pool [5–7]. Furthermore, levels of cfDNA fluctuate several-fold during exercise [8–10] but not as part of circadian rhythm [11, 12]. A comprehensive understanding of cfDNA dynamics in healthy individuals will allow the design of assays that maximise the signal driven by pathological changes, while excluding fluctuations that are part of healthy physiological processes.

Despite the fact that women comprise 50% of the population and most undergo active menstruation generally between the years ~12–50, there has been little investigation of the impact of the menstrual cycle on cfDNA levels. The menstrual cycle involves major remodelling of endometrial tissue. During the follicular phase, stroma, glands and spiral arteries proliferate in the outer functional layer of the endometrium, thickening it from approximately 5.4 mm to 9.2 mm [13]. Without fertilisation and implantation of the ovum, this is eventually followed by menstruation [14], characterised by disintegration of the outer endometrial epithelium layer via apoptosis, fragmentation of glands and loss of adhesion molecules [15–18]. Associated with this is extensive angiogenesis that occurs in the basal endothelium for vascular bed repair [19] as well as inflammation [20]. Leukocytes are known to increase in number and contribute to matrixmetalloproteinases that break down the endometrium [21].

Given the inflammation, apoptosis and the consequential dramatic changes in tissue volume and architecture, we anticipated that during menstruation there would be an increase in total cfDNA, as well as an increase in cfDNA derived from endothelial cells lining the blood vessels within the endometrium. Two previous studies have examined whether cfDNA levels are altered during menstruation, however both studies used serum rather than plasma as the substrate [22, 23]. The majority of cfDNA in serum is derived from leukocytes that lyse during clotting [24, 25], thus serum cfDNA levels do not reflect physiological levels in blood and are highly dependent on sample processing time. To eliminate the possibility that changes in cfDNA were obscured in previous studies by sample artefacts related to serum processing, we quantitated cfDNA fluctuations during the menstrual cycle using plasma samples and with tightly controlled processing protocols.

We also examined whether there was a change in the proportion of the different cell types that contribute to the cfDNA pool during menstruation, specifically, whether there is an increase in endothelial cell-derived cfDNA. Endothelial cells internally line the blood vessels in the body, and are in continuous contact with blood [26]. Despite the large interface between blood and endothelium, initial work suggested that cfDNA from endothelial cells is not present in blood plasma [27]. However, this negative result, based on PCR detection of endothelial-specific unmethylated DNA E-selectin region, was due to the primers used spanning the transcription start site (TSS) of E-selectin. It is now accepted that the TSS of actively transcribed genes is not preserved in cfDNA because it is lacking in nucleosomes that protect it from degradation [28]. Recent work using Illumina methylation profiling showed that endothelial cell DNA actually comprises around 9% of the total cfDNA pool [5]. We utilised the principle of tissue-specific methylation to quantify cfDNA derived from endothelial cells and measure whether it was altered during menstruation. An observed increase would show that menstruation alters the pattern of tissues that contribute to the cfDNA pool in healthy women, which may in turn impact methylation-based diagnostic assays.

## Materials and methods

### Ethics approval and participant recruitment

The recruitment of healthy female volunteers and blood collection was approved by the University of New South Wales Human Research Ethics committee (HC17020). Volunteers were invited to participate via flyers seeking healthy voluneteer blood donors distributed within the University of New South Wales. Researchers were contacted by participants at which point

screening occurred by the exclusion criteria which were pregnancy, lactation, and personal history of cancer. At the scheduled meeting time, which was within working hours, all participants were informed of the study design, written consent was obtained, and blood was collected in a dedicated venpuncture room.

## Participant cohort

Venous blood was collected from each of 40 women twice, once at menstruating (M) and once at non-menstruating (NM) phases of their cycle. Collection began in April 2018 and ceased September 2018. NM samples included both follicular and luteal phases of the cycle. Menstruation status was self-reported and M-phase blood was collected 1 or 2 days after the start of menstruation, when endometrial shedding is profuse. 80% of the sample pairs were collected within the interval of a single menstrual cycle, and no collections were more than three cycles apart (S1 Table). The shortest and longest interval between the NM and M samples from each participant was 7 days and 78 days, respectively. The age range of the participants was between 21 and 49 (median: 29.5, average: 30) years old. The majority of women (33/40) fell within normal body mass index (BMI) range while 2/40 were underweight and 5/40 were overweight. Additionally, 38/40 were non-smokers while 1/40 was a social, on-off smoker for 6 years and 1/40 was a regular smoker for 3 years (S1 Table). Both smoked on average 1 cigarette per day.

## Blood collection

80 mL of peripheral blood was drawn into 8 x 10 mL K2EDTA collection tubes (Becton Dickinson) and processed within 3 hours of collection. The blood was centrifuged at 2500 *xg* for 10 minutes at 4˚C, then the plasma was transferred into a new tube and re-centrifuged at 3500 *xg* for 10 minutes at 4˚C to remove residual contaminating cells. Plasma was stored at -80˚C until cfDNA extraction.

## CfDNA extraction and quantification

A total of 80 cfDNA samples were extracted from 5 mL plasma, with 1 μg carrier RNA, using QIAamp Circulating Nucleic Acid Kit (QIAGEN) as per manufacturer's instructions and eluted in 30 μL of elution buffer. Total cfDNA was quantified using qPCR targeting an ALU repetitive sequence, amplifying a 115 bp product (Forward 5'–CCTGAGGTCAGGAGTTCGAG–3'; Reverse 5'–CCCGAGTAGCTGGGATTACA–3') (henceforth denoted as ALU-115) as well as a 247 bp product (Forward 5'–GTGGCTCACGCCTGTTAATC–3'; Reverse 5'–CAGG CTGGAG TGCAGTGG–3') (henceforth denoted as ALU-247). Serially diluted (1 in 5) commercial human genomic DNA purified from buffy coat (Roche) was used as the standard curve. Each 20 μL PCR reaction contained: 0.01 μL of the eluted cfDNA (1.66 μL of plasma equivalent), 1X PCR Reaction Buffer (Thermo Fisher Scientific), 0.2 mM dNTP Solution Mix (New England Biolabs), 0.06 U/ μL Platinum Taq DNA Polymerase (Thermo Fisher Scientific), 2.5 μM Syto 9 (Thermo Fisher Scientific), 3 mM MgCl$_2$ (Thermo Fisher Scientific), and 0.2 μM each ALU forward and reverse primers (Sigma-Aldrich). The qPCR started with 95˚C for 10 min then for 45 cycles, 95˚C for 30s, 60˚C for 30s and 72˚C for 30s (Thermo Fisher Scientific QuantStudio ViiA 7 Real-Time System). Quantification of total cfDNA was expressed as ng of cfDNA per 1 mL of plasma. The integrity of cfDNA indicated by the ratio of short to long fragments of ALU was quantified by:

$$\text{Ratio of ALU}: \frac{\text{CfDNA concentration (ALU} - 115)}{\text{CfDNA concentration (ALU} - 247)}$$

## Positive and negative controls for methylation specific qPCR

The *in-vitro* Human Methylated and Non-methylated DNA Set (Zymo Research) was used to establish the qPCR conditions that selectively amplified unmethylated CDH5 DNA. Human primary aortic endothelial cells and the hCMEC/D3 blood-brain barrier endothelial cell line (Millipore) were used as positive controls to test the sensitivity and specificity of the unmethylated CDH5 region qPCR primers on biological samples. Cell line and primary cell genomic DNA was extracted using the 'Cultured Cells' protocol of the DNeasy Blood and Tissue Kit (QIAGEN) as per manufacturer's instructions. Genomic DNA was eluted in 40 μL and quantified using nanodrop prior to bisulfite conversion.

## Bisulfite conversion

Bisulfite conversion was performed with Epitect Bisulfite Kit (Qiagen) as per manufacturer's instructions, without carrier RNA. 24 μL cfDNA elution (equivalent to 4 ml of plasma) of the NM and M matched samples was converted using the "Fragmented DNA" protocol and eluted in 30 μL. 1 μg of the *in-vitro* unmethylated and methylated human DNA set, 500 ng human primary aortic genomic DNA (HPA gDNA), and 500 ng blood brain barrier endothelial genomic DNA (BBB gDNA) were all converted using the genomic DNA protocol and eluted in 50 μL each.

## Methylation specific CDH5 primer design and selection

The human genome CDH5 region, encoding VE-cadherin, was selected from the literature as specifically unmethylated in endothelial cells [29]. Primers were designed to selectively amplify unmethylated sequences, with the segment 150 bp upstream to 100 bp downstream of TSS excluded to avoid the nucleosome depleted region. The amplified product was also restricted to less than 100 bp in length to maximise assay sensitivity in cfDNA samples. The resulting primer sequences are Forward 5'-TGTGTTTAAGATGGGAGGGTTT-3'; Reverse 5'-AACC CAACATACCCTCAAAAA -3' and produce a 96 bp size amplicon. Bisulfite converted *in-vitro* human unmethylated (1 ng per reaction) and methylated (1 ng per reaction) genomic DNA was used in a qPCR with varying $MgCl_2$ and temperatures to select optimised conditions that selectively amplify unmethylated DNA, while specifically not amplifying methylated DNA.

## Endothelial cfDNA quantification

Endothelial-derived cfDNA was quantified using the methylation status-specific CDH5 qPCR primers described above against a standard curve generated with bisulfite converted *in-vitro* unmethylated human DNA (Zymo Research) serially diluted 1 in 2. The 20 μL PCR reaction contained: 5 μL of the eluted bisulfite-converted cfDNA (0.667 mL of plasma equivalent), 1X PCR Reaction Buffer (Thermo Fisher Scientific), 0.2 mM dNTP Solution Mix (New England Biolabs), 0.15 U/μL Platinum Taq DNA Polymerase (Thermo Fisher Scientific), 2.5 μM Syto 9 (Thermo Fisher Scientific), 2.5 mM $MgCl_2$ (Thermo Fisher Scientific), and 0.2 μM CDH5 forward and reverse each (Sigma-Aldrich). The qPCR started with 95˚C for 3 min then for 45 cycles, 95˚C for 10s, 62˚C for 20s and 72˚C for 30s (Thermo Fisher Scientific QuantStudio ViiA 7 Real-Time System). The quantification was expressed as ng of endothelial cfDNA per 1 mL of plasma. The relative proportion of the endothelial cfDNA to total cfDNA amount was calculated by:

$$\text{Relative proportion} : \frac{\text{Endothelial cfDNA amount (ng per 1 mL plasma)}}{\text{Total cfDNA amount (ng per 1 mL plasma)}}$$

### Statistical analysis

Statistical analysis was carried out with GraphPad Prism (version 8.4.3). Data are presented as mean with standard deviation. To compare between NM and M samples in total cfDNA, endothelial cfDNA, size ratio, and endothelial cfDNA as a proportion of the total cfDNA, paired, one-tailed, parametric t-test was used. All correlation data used one-tailed Pearson's correlation test. A p value of $< 0.05$ was considered significant.

## Results

### Total cfDNA levels are unaltered by menstruation status

We used qPCR to measure plasma cfDNA levels in matched blood samples from women at M and NM phases of the menstrual cycle. ALU-115 was used to measure total cfDNA, while ALU-247 was used to measure long DNA fragments. We observed no statistically significant difference in the concentration of total cfDNA in both ALU-115 (NM average = 1.79 ± 0.96 ng/mL plasma; M average = 1.68 ± 0.88 ng/mL plasma) and ALU-247 (NM average = 0.66 ± 0.37 ng/mL plasma; M average = 0.64 ± 0.34 ng/mL plasma) at the two phases of the cycle (Fig 1A). CfDNA levels fluctuated up to 7.7-fold between the NM and M blood draws, and we found little to no correlation in cfDNA concentration between the two phases as measured with either ALU-115 or ALU-247 (Fig 1B).

We used the ratio of the ALU-115 and ALU-247 concentration as an indicator of cfDNA size distribution and integrity and found that this also did not change between NM (average = 2.84 ± 0.85) and M (average = 2.69 ± 0.56) phases of the cycle (Fig 2A). Interestingly, we did observe a positive correlation between the cfDNA size ratio of M and NM samples, showing that the integrity of cfDNA is more consistent across time than concentration (Fig 2B).

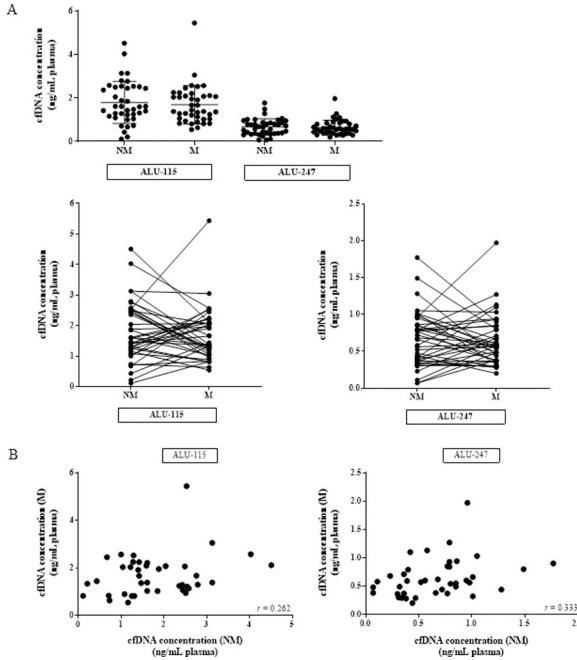

**Fig 1. Total cfDNA does not increase during menstruation.** (A) Total cfDNA quantification in 40 healthy women at NM and M phases by qPCR, expressed in ng of cfDNA per 1 mL of plasma (ALU-115 p = 0.273; ALU-247 p = 0.385). (B) Scatter plot of NM ALU-115 and M ALU-115 (r = 0.262; p = 0.0516) as well as NM ALU-247 and M ALU-247 (r = 0.333; p = 0.0179).

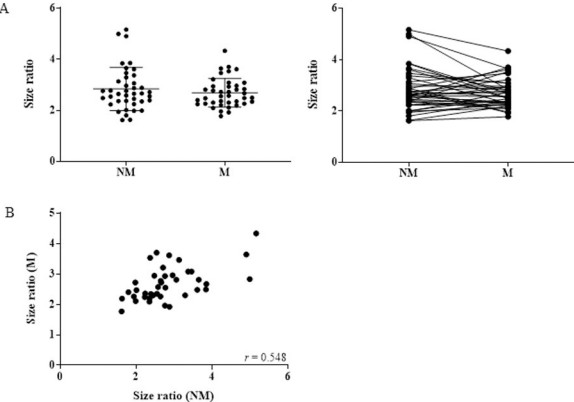

**Fig 2. Integrity of cfDNA does not change during menstruation.** (A) CfDNA integrity assessed via the ALU-115/ALU-247 size ratio in 40 healthy women at NM and M phases (p = 0.0931). (B) Scatter plot of size ratio at NM and M phases (r = 0.548; p = 0.0001).

## Endothelial specific DNA primer design and validation

The CDH5 region downstream of the TSS was found to be suitable for unmethylated DNA-specific primer design. The primers contain a total of 4 mismatches between methylated and unmethylated bisulfite-converted DNA (S1A Fig). With the same amount of bisulfite-converted genomic DNA input (1 ng), the primer was able to selectively amplify the *in-vitro* unmethylated genomic DNA but not the methylated genomic DNA, which performed the same as the no template control (NTC) (S1B–S1C Fig). The primer was further validated by amplifying serially diluted bisulfite-converted HPA gDNA and BBB gDNA. The lower limit of detection was 111 pg and 125 pg in HPA and BBB dilutions, respectively (S2 Fig).

## Endothelial cell-derived cfDNA levels are unaltered by menstruation status

While we did observe considerable increases or decreases in individual matched samples, the direction of the change was not consistent across the whole cohort and, like the total cfDNA, no overall change in average endothelial-derived cfDNA concentration was observed at NM (average: 1.01 ± 0.57 ng/mL plasma) compared to M phases (average: 1.07 ± 0.57 ng/mL plasma) in 40 matched samples (Fig 3A).

Similarly, no significant difference was observed when the concentration was adjusted against the total cfDNA ALU-115 concentration to express endothelial cfDNA relative to the total (Fig 3B). There was a statistically significant (p = 0.0052) positive correlation in endothelial cfDNA between the NM and M phases, showing that the proportion of endothelial cfDNA, is consistent across time (Fig 3C).

## Discussion

We compared a range of cfDNA parameters in matched plasma samples from women at menstruating and non-menstruating phases of their cycle. Specifically, we measured total cfDNA concentration and size integrity, and the relative amount of cfDNA contributed by endothelial cells.

With 40 pairs of matched samples, our study has 90% power to detect a 30% increase, and >95% power to detect an increase of 35% and over. We found no change in total cfDNA concentration during menstruation, and no change in the size distribution as measured by the ratio of two different amplicon sizes, ALU-115 and ALU-247. The ALU-247 concentration

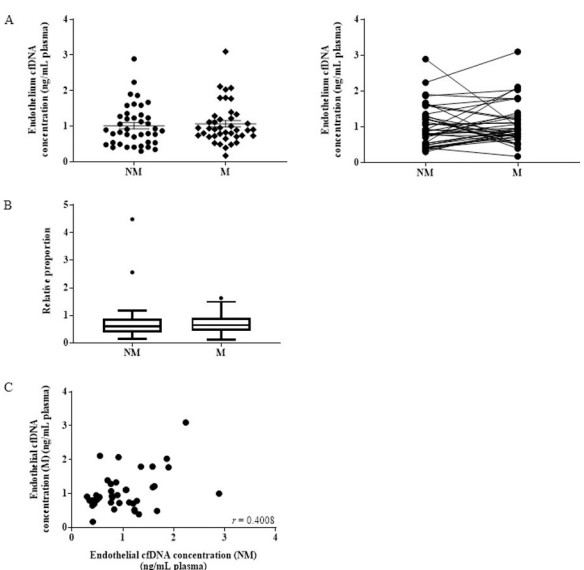

**Fig 3. Endothelial cell-derived cfDNA does not increase during menstruation.** (A) Measurement of endothelial cfDNA quantification by qPCR in 40 healthy women at NM and M phases, expressed in ng of cfDNA per 1 mL of plasma (p = 0.301). (B) Box plot graph showing the proportion of endothelial cfDNA compared to total cfDNA at NM and M phases (p = 0.3686). (C) Scatter plot of endothelial cfDNA at NM and M phases (r = 0.4008; p = 0.0052).

magnitude was less than the ALU-115, as expected since this amplicon is too long to detect the shorter cfDNA population. We also found no change in the endothelial-derived cfDNA, for both amount and when expressed relative to the total.

The ladder pattern of cfDNA when subjected to gel electrophoresis has prompted the hypothesis that this DNA is derived from apoptotic cells and enters the blood stream as debris that has bypassed macrophage clean-up mechanisms [30]. In this scenario, we would expect menstruation to raise cfDNA levels, as it is a process that involves extensive apoptosis and tissue remodelling [16–18]. The lack of increase we observed suggests that menstruation involves uniquely efficient removal of apoptotic cells, possibly aided by the fact that tissue breakdown products can be shed into the uterine cavity during menses [23]. An alternative explanation is that in healthy individuals apoptotic debris is always very efficiently removed, and specific processes unrelated to apoptosis, for example erythrocyte enucleation [4, 5] and NETosis [31], are responsible for the bulk of cfDNA. The latter is consistent with erythrocyte progenitors having been shown to contribute approximately 27–30% of the cfDNA total [4, 5], with granulocytes, which are potentially linked to NETosis [31], contributing another 32% [5]. If apoptosis is not the main source of cfDNA from healthy cells, it may account for the different size distributions of tumour and healthy cfDNA that have been reported [32], as the two would be released by different pathways.

It is also possible that we did not observe an increase in cfDNA because menstruation leads to DNA fragmentation to a size that cannot be amplified by the ALU-115 and the endothelial cell specific primers, which create 115 bp and 96 bp products respectively. It has been reported that cancer [33], graft transplants [34], stroke [35] as well as non-disease settings, such as pregnancy [36, 37], can lead to more pronounced fragmentation of cfDNA. In the cancer context, this fragmentation can shorten cfDNA, compared to healthy controls, in both total cfDNA pool [38–40] and in tumour-specific fragments, from ≥100 bp down to 57–85 bp [32, 41–43].

In contrast to the variation in total cfDNA and endothelial cfDNA concentrations, we observed some consistency in cfDNA size ratios, and in endothelial cfDNA between M and

NM phase samples (r = 0.548, p = 0.0001 and r = 0.4008, p = 0.0052, respectively). This suggests that the cleavage rates and cell-sources of cfDNA within healthy individuals are somewhat constant over time, and less susceptible to physiological fluctuations.

An association between endothelial cells and circulating nucleic acids is not novel. The presence of endothelial cell mRNA is significantly increased along with total cfDNA in burn patients [44]. This cfDNA elevation is also observed in major trauma [45] and cardiac surgery with cardiopulmonary bypass patients [46] with a positive correlation with endothelial damage-specific markers.

A limitation of this study is that non-menstruating samples were not differentiated between ovulation, luteal and follicular phases. It is possible that these influence cfDNA levels, however, our hypothesis was that the changes that occur during menstruation will increase cfDNA levels, and this was not supported by the data. Measuring changes during non-menstruating phases was beyond our scope to of our study. Furthermore, we did not exclude women with current use of oral contraceptive, and while we do not anticipate a cofounding effect from this, this has yet to be investigated in the literature.

Although in our cohort we did not observe cfDNA changes relating to menstrual status, individual cfDNA levels did fluctuate over time. We found up to 7.7-fold variation within a single individual, and no correlation between the two consecutive measurements across the whole cohort. CfDNA levels are known to be increased by exercise [47], and it is possible that different levels of physical activity prior to blood donation contributed to the variation we observed. However, low intensity exercise is not sufficient to measurably change concentration [48], and even following intense exercise cfDNA has been reported to return to baseline within about 1 hour [10, 49, 50], so this is not likely to account for all the changes. However, based on published literature we do note that the vast majority of research into cfDNA and exercise has been conducted on men and therefore mechanisms of cfDNA release and absorption unique to women may be unidentified [51]. Our data highlight the lack of knowledge of factors that drive cfDNA fluctuations in healthy individuals.

## Conclusion

Menstruation has little impact on the amount, size distribution and cell type contributions of cfDNA in blood plasma, suggesting that highly efficient clearance mechanisms operate during endometrial tissue remodelling. In the context of cfDNA biomarker development and cancer screening tests, menstrual status at the time of diagnostic blood collection should not impact on quantity of total cfDNA obtained. However, more research is required into cfDNA release and absorption in healthy individuals.

## Supporting information

**S1 Table. Demographic of 40 healthy female volunteers.**
(DOCX)

**S1 Fig. CHD5 primers are specific for unmethylated DNA.** (A) A schematic of primer location in the CDH5 region spanning from +122 bp to +218 bp. CpG mismatches are shown as •. The arrows indicate forward and reverse primers. qPCR amplification (B) and melt curve (C) plot of CDH5 primer selectivity and specificity in 1 ng of *in-vitro* unmethylated (green) and methylated DNA (red) set with NTC (black) as control.
(TIF)

**S2 Fig. Reproducible quantification of unmethylated CDH5 in human primary endothelial cells.** qPCR amplification plot of CDH5 in (A) human aortic endothelial cells (3 ng and 111

pg) and (B) blood brain barrier endothelial cells (2 ng and 125 pg) as primer validation.
(TIF)

## Acknowledgments

We would like to acknowledge and thank the volunteers for their blood donation. Furthermore, we thank Dr. Rob Rapkins, Dr. Lindsay Wu and Catherine Li for providing the primary endothelial cells as well as Dr. Nancy Briggs for performing the power calculation and verifying the statistics.

## Author Contributions

**Conceptualization:** Nicole Laurencia Yuwono, Caroline Elizabeth Ford, Kristina Warton.

**Data curation:** Nicole Laurencia Yuwono.

**Formal analysis:** Nicole Laurencia Yuwono, Caroline Elizabeth Ford.

**Funding acquisition:** Nicole Laurencia Yuwono, Caroline Elizabeth Ford, Kristina Warton.

**Investigation:** Nicole Laurencia Yuwono, Claire Elizabeth Henry.

**Methodology:** Nicole Laurencia Yuwono, Claire Elizabeth Henry, Caroline Elizabeth Ford, Kristina Warton.

**Supervision:** Caroline Elizabeth Ford, Kristina Warton.

**Writing – original draft:** Nicole Laurencia Yuwono, Caroline Elizabeth Ford, Kristina Warton.

**Writing – review & editing:** Nicole Laurencia Yuwono, Claire Elizabeth Henry, Caroline Elizabeth Ford, Kristina Warton.

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
