## [Decision Letter · Decision Letter 0]

8 Mar 2021

PONE-D-20-36096

Total and Endothelial Cell-Derived Cell-Free DNA in Blood Plasma Does Not Change During Menstruation

PLOS ONE

Dear Dr. Warton,

First of all, please accept my sincere apologies for the long time required for the review. Finding adequate reviewers was particularly difficoult.

After careful consideration, we feel that it has merit but does not fully meet PLOS ONE’s publication criteria as it currently stands. Therefore, we invite you to submit a revised version of the manuscript that addresses the points raised during the review process by both reviewers.

We look forward to receiving your revised manuscript.

Kind regards,

Francesco Bertolini, MD, PhD

Academic Editor

PLOS ONE

Journal Requirements:

2. In your Methods section, please provide additional information about the participant recruitment method and the demographic details of your participants. Please ensure you have provided sufficient details to replicate the analyses such as: a) the recruitment date range (month and year), b) a table of relevant baseline demographics and characteristics c) a description of how participants were recruited, and d) descriptions of where participants were recruited and where the research took place.

"I have read the journal's policy and the authors of this manuscript have the following competing interests: KW holds stock in Guardant Health, Exact Sciences and Epigenomics AG. No other authors have competing interests. "

Reviewers' comments:

Reviewer's Responses to Questions

**Comments to the Author**

1. Is the manuscript technically sound, and do the data support the conclusions?

Reviewer #1: Yes

Reviewer #2: Partly

2. Has the statistical analysis been performed appropriately and rigorously? 

Reviewer #1: Yes

Reviewer #2: Yes

3. Have the authors made all data underlying the findings in their manuscript fully available?

Reviewer #1: Yes

Reviewer #2: Yes

4. Is the manuscript presented in an intelligible fashion and written in standard English?

Reviewer #1: Yes

Reviewer #2: Yes

5. Review Comments to the Author

Reviewer #1: This manuscript technically sounds and the data reported support the central hypothesis. I have absolutely no objections regarding the statistical analysis performed in this article; the data supporting the findings are fully available. I've read the article with attention and one thing that doesn't fully convince me is the number of volunteers enrolled in the study(40). As I said before, the results are clear, but i'd probably use more volunteers in order to obtain stronger evidences.

Reviewer #2: In the presented work, authors investigated modulation of cell-free DNA (cfDNA) in blood of 40 healthy women comparing menstruating (M) and non-menstruating (NM) days concluding that there is no difference between these two periods in cfDNA and in endothelial-specific cfDNA.

Investigating the cfDNA in healthy subjects clarly represent a relevant issue to generate new insoight on cfDNA but sample size and sample collection are two critical points. The use of cfDNA, as reported within the manuscript , could be considered as promising tool in biomarker discovery; therefore there are many criticisms to be addressed.

Authors used only 40 healthy subjects, without multiple sample collection during time and only discrimination between M and NM period is not sufficient to support the overall conclusions of the manuscript.

Based on what parameters the enrolled women were considered healthy?

Considering that the menstrual cycle is controlled by hormonal changes and that can be identified a follicular phase, ovulation and luteal phase, have authors considered these 3 phases as possible influencing factors?

Blood sample collection during the NM phase is comparable among the studied subjects?

In addition, authors reported, in table 1S, that some enrolled women were undergoing oral contraceptive therapy (11/40, 27.5%). Can this type of therapy influence cfDNA? Can authors exclude that cfDNA can vary in relation to hormonal fluctuation or oral contraceptive therapy?

Can authors exclude that the absence of differences in cfDNA that have been reported is not related to different timing of blood collection during NM period?

Age of enrolled women is from 21 to 49 years old and considering that biomarkers can be modified/modulated during age, can this large age range influence results? Is it possible that age is a confounding factor? Or can authors exclude this hypothesis?

6. PLOS authors have the option to publish the peer review history of their article (what does this mean?). If published, this will include your full peer review and any attached files.

Reviewer #1: No

Reviewer #2: No

---

## [Author Response · Author response to Decision Letter 0]

20 Mar 2021

Dear reviewers, 

Re: ‘Total and Endothelial Cell-Derived Cell-Free DNA in Blood Plasma Does Not Change During Menstruation’ (PONE-D-20-36096)

We would like to thank the reviewers for their feedback on our manuscript. We appreciate the effort that has been put into its evaluation. 

In response to reviewer 1:

Re: I've read the article with attention and one thing that doesn't fully convince me is the number of volunteers enrolled in the study (40). As I said before, the results are clear, but i'd probably use more volunteers in order to obtain stronger evidence.

In view of the feedback from Reviewer 1 on cohort size (also raised by Reviewer 2), we re-visited the statistical power of our study. The numbers reported in our original manuscript (i.e. “greater than 80% power to detect a 40% concentration increase”, p12, lines 302-303), were based on calculations from an initial recruitment target of 36 subjects. This recruitment target was exceeded in our study, as we collected ~10% extra samples in case some were hemolyzed or otherwise unusable, but all plasma samples turned out to be good quality and included in the analysis.

We have updated the text of our manuscript to reflect the actual cohort size and re-calculated power. With the current cohort of 40 paired samples and a standard deviation of 0.63, our study had 80% power to detect a 25% increase, 90% power to detect a 30% increase, and >95% power to detect an increase of 35% and above. We therefore believe our sample size is sufficient to see an increase if the biological phenomenon occurs to an appreciable extent. We have made an amendment in the discussion section based on this (page 12 lines 304 -305). 

We agree with the reviewer that a larger sample size would create even stronger confidence in the conclusion of the study.

In response to reviewer 2: 

Re: Authors used only 40 healthy subjects, without multiple sample collection during time and only discrimination between M and NM period is not sufficient to support the overall conclusions of the manuscript. 

We carried out 2 blood collections from each subject – menstruating and non-menstruating. We agree that this does not address the possibility of cfDNA variation during luteal and follicular phases, and at ovulation. However, given that the most rapid and dramatic structural changes occur in endometrium at menstruation, we focused on this phase for our study, and we limit our conclusion to stating that there was no change in cfDNA during menstruation (line 305-306, page 12 “we found no change in total cfDNA concentration during menstruation”). 

Our manuscript does not conclude that there is no change in cfDNA levels between, for example, the luteal and follicular phases, and we have amended our discussion to emphasize that there may still be a differences between luteal, follicular and ovulation phases that our study was not designed to detect (lines 356-362, p14).

With regards to the number of subjects, we note that our study benefited from paired sample analysis, which helped us achieve a power of greater than 95% to detect a 35% increase in cfDNA parameters (please see reply to Reviewer 1 above). 

We also note that we did not observe a trend towards change, which lacked only sufficient cohort size to be statistically significant. The values in cfDNA concentration were extremely similar (1.79 ng/mL and 1.68 ng/mL), and it would require a very large number of participants to measure an effect of this size with adequate power. We do feel that this would be a worthwhile undertaking in the context of a much larger study, but is beyond the scope of the current report. 

Re: Based on what parameters the enrolled women were considered healthy?

Participants were considered healthy based on self-reporting at the time of recruitment. Upon blood collection, participants were required to fill out a questionnaire in which they had to state any medical history thereby giving them an opportunity to inform the researcher if they have any past or current comorbidities. None of the participants had any existing conditions, including but not limited to cold or flu, at the time of both NM and M phase collection, or any personal history of cancer. We have expanded the description of participant recruitment to provide more information (p5, lines 111-119). 

Re: Considering that the menstrual cycle is controlled by hormonal changes and that can be identified a follicular phase, ovulation and luteal phase, have authors considered these 3 phases as possible influencing factors?

Collection in the three phases is a great recommendation and would be an exciting study to do in plasma. We would have liked to collect blood from the luteal and follicular phase, and at ovulation for comparison with cfDNA levels during menstruation, however, this would have involved monitoring the 40 subjects for timing of ovulation. Such a study design would involve an immense amount of tracking as we would need to subject the participants to multiple blood tests to accurately measure the phases. Considering the variability of cycle time in each individual, this would be beyond the scope of our study. 

Since our hypothesis was that menstruation, characterized by shedding of endometrial tissue, will cause endothelial cfDNA to be increased, the state of shedding (M) vs non-shedding (NM) was adequate within the context of the question we posed. 

We have amended our discussion to incorporate this comment as a limitation on page 14 lines 356 - 362. as follows: 

A limitation of this study is that non-menstruating samples were not differentiated between ovulation, luteal and follicular phases. It is possible that these influence cfDNA levels, however, our hypothesis was that the changes that occur during menstruation will increase cfDNA levels, and this was not supported by the data. Measuring changes during non-menstruating phases was beyond our scope to of our study. 

Re: Blood sample collection during the NM phase is comparable among the studied subjects?

Can authors exclude that the absence of differences in cfDNA that have been reported is not related to different timing of blood collection during NM period?

We aimed to collect NM samples 14-18 days after the beginning of menstruation and within a window of 1 cycle. As mentioned in the previous response, our study was designed to observe the impact of endometrial shedding during menses. If we designed the study as to collect at a specific non-menstruating phase we would have had to perform an immense amount of tracking via additional blood tests in each participant which is not feasible in the scope of our study. A change in cfDNA levels between different phases if the NM period is possible, and a worthwhile topic for future study, however, it seems less likely than a change during menstruation. 

Re: In addition, authors reported, in table 1S, that some enrolled women were undergoing oral contraceptive therapy (11/40, 27.5%). Can this type of therapy influence cfDNA? Can authors exclude that cfDNA can vary in relation to hormonal fluctuation or oral contraceptive therapy?

To the best of our knowledge, the effect of oral contraception on cfDNA is yet to be investigated. We stratified our 40 matched cohort into oral contraception non-user (29/40) and user (11/40) and performed a two-way ANOVA analysis after the stratification. The analysis showed no significant effect from the use of oral contraception.

As above, we note that we were aiming to measure any change in cfDNA caused by endometrial shedding during menses, and as such, non-menstruating women, either using or not using oral contraception, were an adequate control. We appreciate that if we were monitoring changes during the luteal and follicular phases of the cycle, including women using oral contraception in the control group would not have been appropriate. 

We have added to following comment to the discussion to highlight this as a limitation of our study (p.14, lines 361-363):

Furthermore, we did not exclude women with current use of oral contraceptive, and while we do not anticipate a cofounding effect from this, this has yet to be investigated in the literature. 

Re: Age of enrolled women is from 21 to 49 years old and considering that biomarkers can be modified/modulated during age, can this large age range influence results? Is it possible that age is a confounding factor? Or can authors exclude this hypothesis?

Since paired samples were collected from subjects, the menstruating and non-menstruating samples are exactly age-matched, and age is not likely to be confounding factor. Additionally, we performed a correlation analysis between the cfDNA amount and age of the participants tested in our study and there was none. 

In response to journal requirement

Comment #1 PLOS ONE’s style requirements.

We have reviewed and amended our manuscript to the requirements of PLOS ONE. If there are any further formatting and layout changes required, please do not hesitate to let us know. 

Comment #2 Better detail on participant recruitment. 

We have included more detail regarding the recruitment process (how, where and when) in the methods section (page 5, lines 111-119) as well as expanding participant demographic characteristics in Supplementary Table 1 and its description in the methods section (pages 5-6, lines 131-142). 

Comment #3 Update Competing Interest statement. 

Please see below for the updated Competing Interest statement:

I have read the journal's policy and the authors of this manuscript have the following competing interests: KW holds stock in Guardant Health, Exact Sciences and Epigenomics AG. No other authors have competing interests. This does not alter our adherence to PLOS ONE policies on sharing data and materials. There are no restrictions on sharing of data and/or materials from this publication. 

We thank the editor and reviewers for their time in assessing our response to the comments. Please do not hesitate to let us know if there are anything further that we can do. 

Sincerely, 

Dr. Kristina Warton 

Corresponding author

k.warton@unsw.edu.au

(02) 9385 1439

Gynaecological Cancer Research Group 

School of Women’s and Children’s Health 

Faculty of Medicine 

University of New South Wales 

Sydney, Australia

---

## [Decision Letter · Decision Letter 1]

12 Apr 2021

Total and Endothelial Cell-Derived Cell-Free DNA in Blood Plasma Does Not Change During Menstruation

PONE-D-20-36096R1

Dear Dr. Warton,

We’re pleased to inform you that your manuscript has been judged scientifically suitable for publication and will be formally accepted for publication once it meets all outstanding technical requirements.

Kind regards,

Francesco Bertolini, MD, PhD

Academic Editor

PLOS ONE

Additional Editor Comments (optional):

Reviewers' comments:

Reviewer's Responses to Questions

**Comments to the Author**

1. If the authors have adequately addressed your comments raised in a previous round of review and you feel that this manuscript is now acceptable for publication, you may indicate that here to bypass the “Comments to the Author” section, enter your conflict of interest statement in the “Confidential to Editor” section, and submit your "Accept" recommendation.

Reviewer #1: All comments have been addressed

Reviewer #2: All comments have been addressed

2. Is the manuscript technically sound, and do the data support the conclusions?

Reviewer #1: Yes

Reviewer #2: Yes

3. Has the statistical analysis been performed appropriately and rigorously? 

Reviewer #1: I Don't Know

Reviewer #2: Yes

4. Have the authors made all data underlying the findings in their manuscript fully available?

Reviewer #1: Yes

Reviewer #2: Yes

5. Is the manuscript presented in an intelligible fashion and written in standard English?

Reviewer #1: Yes

Reviewer #2: Yes

6. Review Comments to the Author

Reviewer #1: (No Response)

Reviewer #2: The atuthors provided a detaile rebuttal letter trying to address, to their best the specific comments by the reviewer. The major concern was specifically addressed on the statistical analysis on the provided sample size and satisfactory responsed have been provided and sustained.

The authors also better stressed on the limintayion of their study.

I would consider the manuscript to be accepted to the journal.

7. PLOS authors have the option to publish the peer review history of their article (what does this mean?). If published, this will include your full peer review and any attached files.

Reviewer #1: No

Reviewer #2: No

---

## [Editor Report · Acceptance letter]

15 Apr 2021

PONE-D-20-36096R1 

Total and Endothelial Cell-Derived Cell-Free DNA in Blood Plasma Does Not Change During Menstruation 

Dear Dr. Warton:

I'm pleased to inform you that your manuscript has been deemed suitable for publication in PLOS ONE. Congratulations! Your manuscript is now with our production department. 

Kind regards, 

on behalf of

Dr. Francesco Bertolini 

Academic Editor

PLOS ONE